# Successful Liver Transplantation in Two Polish Brothers with Transaldolase Deficiency

**DOI:** 10.3390/children8090746

**Published:** 2021-08-29

**Authors:** Marek Stefanowicz, Maria Janowska, Joanna Pawłowska, Anna Tylki-Szymańska, Adam Kowalski, Marek Szymczak, Piotr Kaliciński, Irena Jankowska

**Affiliations:** 1Department of Pediatric Surgery and Organ Transplantation, The Children’s Memorial Health Institute, 04-730 Warsaw, Poland; m.stefanowicz@ipczd.pl (M.S.); a.kowalski@ipczd.pl (A.K.); m.szymczak@ipczd.pl (M.S.); p.kalicinski@ipczd.pl (P.K.); 2ERN Transplant Child, 28020 Madrid, Spain; j.pawlowska@ipczd.pl; 3Department of Gastroenterology, Hepatology, Nutritional Disorders and Pediatrics, The Children’s Memorial Health Institute, 04-730 Warsaw, Poland; i.jankowska@ipczd.pl; 4Department of Metabolic Diseases, The Children’s Memorial Health Institute, 04-730 Warsaw, Poland; A.Tylki@ipczd.pl

**Keywords:** transaldolase deficiency, TALDO, liver transplantation

## Abstract

Transaldolase deficiency (TALDO; OMIM 606003) is a rare inborn autosomal-recessive error of the pentose phosphate pathway. It is an early-onset multisystem disease with dysmorphic features, anaemia, coagulopathy, thrombocytopenia, tubulopathy, hepatosplenomegaly and end-stage liver disease. We present a case of two Polish brothers, born to consanguineous parents, with early-onset TALDO. The dominant feature of disease was an early severe liver injury, with subsequent renal tubulopathy. Nodular liver fibrosis developed in the course of the underlying disease. The older brother presented stable liver function, however, he was qualified for deceased donor liver transplantation (DDLT) because of a liver tumour and suspicion of hepatocarcinoma. The boy was transplanted at the age of 14. The younger brother was qualified for DDLT due to end-stage liver disease and transplanted at the age of 11. Currently, both our patients are alive and in a good condition with normal graft function 23 and 20 months after DDLT respectively. Liver transplantation can be a therapeutic option in TALDO and should be considered in patients with coexisting severe chronic and end-stage liver disease. Long term follow-up is necessary to assess the impact of liver transplantation for quality of life, survival time and the course of the disease.

## 1. Introduction

Transaldolase deficiency (TALDO; OMIM 606003) is a rare inborn autosomal-recessive error of the pentose phosphate pathway. The effect of transaldolase deficiency is the accumulation of polyols—arabitol, erythritol, ribitol, sedoheptitol, perseitol and seven- carbon sugars—sedoheptulose, mannoheptulose and phosphosedoheptulose. Transaldolase deficiency has varied presentation including end-stage liver disease, renal tubular dysfunction, coagulopathy, anaemia, thrombocytopenia, congenital cardiac abnormalities and hormonal disorders. Bone mineralization disorders and short stature are found in patients with TALDO. They have characteristic skin with cutis laxa, a visible vascular network, telangiectasia and a tendency to form cavernous hemangiomas. Psychomotor development is normal [1]. The diagnosis of the disease is based on the identification of urinary excretion and the accumulation of polyols and sugars in the serum. Confirmation of the diagnosis is the assessment of transaldolase activity in peripheral blood leukocytes or cultured fibroblasts or a molecular test (sequence analysis of the *Taldo1* gene) [2]. There is no effective treatment for TALDO.It has been suggested that liver transplantation can be a therapeutic option in patients with transaldolase deficiency and end-stage liver disease. Only two cases of liver transplantation in a 1-year-old children have been reported in the literature so far [3,4]. We present another two patients, siblings, with TALDO deficiency and cirrhosis, who underwent liver transplantation in our centre.

## 2. Materials and Methods

We describe two patients, brothers, born to healthy consanguineous Polish parents. The TALDO disease manifested early in both patients. Prenatal period of Patient I and Patient II were complicated by an excessive weight gain in the mother and atypical enlargement of the placenta. Intrauterine growth retardation was observed in both. From the neonatal or early childhood period, hepatosplenomegaly with coagulopathy, hypoalbuminemia, and elevated transaminase activity were diagnosed. No dysmorphic features were observed. The characteristic thin skin with a visible network of vessels (both patients) and cavernous hemangiomas (Patient II) was noticed. Growth retardation was observed in both brothers, however, intellectual development was normal. Suspicion of the disease in Patient I was made on the basis of GCMS and Tandem MS tests and confirmed by genetic examination at 3 years of age. Molecular analysis revealed a presumably homozygous known mutation c.575G>A in exon 5 TALDO1. In Patient II, molecular analysis at 6 months of age revealed the same genotype as in his brother. Additionally, the concentration of polyols and sugars was increased in fluids.

The pretransplant clinical course and medical characteristics were also previously described by Tylki-Szymańskaet. al [5] and Lipiński et al [6]. The details on pretransplant clinical course of each patient are presented in Table 1.

## 3. Results

### 3.1. Patient I

From infancy, the older brother presented a tendency to bleedings with coagulopathy, deficiency of factors XI and XII, haematological abnormalities including anaemia and thrombocytopenia. End-stage liver disease symptoms occurred in the first year of life with hepatosplenomegaly, elevated aminotransferases and cholestasis. The nodular liver fibrosis was detected by ultrasonography and oesophageal varices were revealed by esophagogastroscopy.

Progression of liver damage led to liver decompensation at the age of 5 yrs, with massive ascites and peripheral oedema (scrotum and eyelids). Laboratory tests revealed worsening synthetic function of the liver with hypoalbuminemia, coagulopathy and hyperaminotransferasemia. Additionally, the patient developed renal dysfunction with tubulopathy. Two years later severe bleeding from oesophagal varices led to hypotension and hypovolemic shock resulting in a coma. During hospitalization the boy has developed spontaneous bacterial peritonitis. He was discharged after long-term treatment for palliative care. At the age of 11, after multidisciplinary discussion, the boy was listed for LT with PELD 7, however, one year later, he was suspended from the list because of stable liver function. Finally, at the age of 14, even though the boy still presented stable liver function, he was activated again due to the suspicion of HCC based on an ultrasound and computed tomography where large nodular features of the liver were revealed.

#### 3.1.1. Liver Transplantation

The boy underwent orthotopic liver transplantation using standard technique at age of 14 yrs (3 months after placing on the waiting list), from a deceased liver donor, matched in blood group (A to A group) (Table 2). For biliary tract reconstruction the Roux-en-Y hepaticojejunostomy was performed. Cold ischemia time was 519 min. There were no any surgical complications after liver transplantation. The episode of a disorientation was noticed, but neurological examinations (including CNS imaging, EEG) did not confirm the organic and functional pathology of CNS. Thereafter, no other neurological symptoms were observed. The histopathological assessment of the explanted native liver did not confirm features of hepatocellular cancer. The features of nodular cirrhosis and secondary degenerative changes were described.

Ultrasound examination of the transplanted liver confirmed good vascular flows with slightly accelerated flow through the hepatic artery (HA Vmax 80 cm/s, RI 0,51) and acceleration in the portal vein (PV Vmax110 cm/s). Follow-up computed tomography scans did not confirm stenosis in the portal vein anastomosis (8/6.5/10 mm).

While checking kidney functions, persistent proteinuria and haematuria, elevated cystatin C and creatinine, with normal urea and albumins levels and correct complement concentration have been revealed (GFR 56.99 mL/min). Ultrasound examination revealed increased echogenicity of the renal parenchyma with a blurred cortical-medullar differentiation. The patient did not require supportive nephrological treatment beyond nephroprotection by ACE-I (Ramipril), but constant monitoring of kidney function was recommended.

Immunosuppressive treatment included tacrolimus (0.2 mg/kg) and mycophenolate mofetil (MMF) (20 mg/kg).Due to persistent thrombocytopenia in the postoperative period, MMF was temporarily discontinued and prednisone was included in the therapy (in decreasing doses and discontinued 3 weeks after discharge home). Due to impaired renal function, the tacrolimus concentration of 6–8 ng/mL was recommended.

The boy was discharged from the Transplant Unit on 19th day with good liver function and normal aminotransferases, with slightly elevated GGTP, normal bilirubin concentration and coagulogram. Mild anaemia and thrombocytopenia were noticed.

#### 3.1.2. Post LT Follow-Up

The patient remains under the care of the Transplant Outpatient Clinic with 2 years follow-up after liver transplantation. Currently, a 16-years-old boy, present normal liver function tests: transaminase activity, GGTP and bilirubin concentration and no coagulopathy. The albumin concentration remains within normal range. Hepatic flows are regular. Current immunosuppression consist of tacrolimus (0.14 mg/kg, concentrations 5–7 ng/mL) and almost completely reduced dose ofMMF (7 mg/kg).The renal function is still impaired but stable with slightly elevated cystatin and creatinine concentrations. He receives nephroprotective treatment with Ramipril. The blood tests still show slight anaemia and intermittent slight leukopenia with normal platelet level.

Despite normal liver function, the patient has grown only 7 cm since transplantation. Both body weight and height remain well below 3 pc (Table 3). For this reason, endocrine diagnostics was started.

### 3.2. Patient II

In Patient II, the younger brother of Patient I, bleeding diathesis with factor XI and XII deficiency, with haematological disorders (anaemia, thrombocytopenia) and elevated transaminase activity was observed from the neonatal period. In the 5th month of age, due to family screening, the diagnosis for TALDO was performed. It was made on the basis of a molecular test showing the same mutation as in Patient I.

At 4 years of age, the patient developed hepatic decompensation with massive ascites, scrotum oedema, coagulation disorders, hypoalbuminemia and cholestasis. Hepatosplenomegaly was described by ultrasound examination. The gastroscopy showed first-degree oesophagal varices. Features of tubulopathy appeared. Treatment with Ramipril was started.

The boy was qualified for liver transplantation by the decision of the multidisciplinary council at the age of 8. The PELD was 12. Due to the stable liver function, it was decided to suspend the patient from the waiting list, similarly to his older brother, but 3 years later it was decided to activate the patient on the liver transplant waiting list due to the gradual deterioration of the liver function (Table 1).

#### 3.2.1. Liver Transplantation

The boy was transplanted from a deceased liver donor, matched in blood group (A to A group), 4 months later at 11 years of age (Table 2). He underwent orthotopic whole liver transplantation using standard technique with duct to duct reconstruction of biliary tract (cold ischemia time was 350 min).On 7 postoperative day patient required surgical intervention due to intraabdominal bleeding. Graft function was good.

Standard initial immunosuppression treatment was administered with tacrolimus and MMF, but because of thrombocytopenia (28 k/uL) and leucopoenia (2 k/uL), MMF was stopped and prednisone therapy was started, but it did not result in the expected improvement in blood tests. Tacrolimus was replaced by cyclosporine (CsA) six days after LT and triple drug therapy was continued (CsA, MMF and prednisone 0.25 mg/kg). The blood morphology improved gradually.

In the postoperative course patient developed some complications. On 10thpostLT day the patient presented a short epileptic episode with an increase in blood pressure. Bleeding into CNS and epilepsy were excluded by the neurological and radiological examinations. Treatment with labetalol was introduced then modified with captopril and increased doses of Polpril due to hypertension. Twenty days after LT activity of aminotransferases increased 3 times normal (ALT 129 U/L, AST 135 U/L) and PCR test revealed CMV DNA copies in the blood (2270 copies per ml) despite the prophylactic use of valganciclovir.The boy was treated with intravenous ganciclovir with normalisation of aminotransferases activity and elimination of CMV DNA, but graft function decreased again one month after transplantation. Clinical diagnosis of acute rejection was followed by the empiric therapy with 6 pulses of methylprednisolone in dosage 10 mg/kg with rapid improvement of the laboratory tests. Tacrolimus was introduced again instead of cyclosporine.

Ultrasound examination detected accelerated flow through the portal anastomosis (PV V max 37/112/79 cm/s). The flow through the hepatic artery was normal (HA Vmax 27 cm/s). The stenosis in the portal vein was excluded in the subsequent follow-up ultrasound examinations.

The patient was discharged home 48 days after LTwith good liver function.

#### 3.2.2. Post LT Follow-Up

Actual postLT follow-up is almost 2 years. At 2 months postLTcheck up increased aminotransferases were observed, but infectious or immunological causes were excluded by viral tests and liver biopsy. MMF was discontinued 6 months after LT because of persistent leucopenia and neutropenia(WBC 1.81 k/uL, neutrophils 0.16 k/uL) with increase of white blood cells level.

Currently, a 13-year-old patient remains under the care of Transplant Outpatient Clinic. He is in good general condition, liver function is preserved with normal transaminase, GGTP activity, normal bilirubin and albumin concentration and no coagulopathy. Immunosuppression consist of tacrolimus (0.17 mg/kg, trough concentrations 5–7 ng/mL) and a reduced dose of prednisone (0.1 mg/kg). Hepatic flows are good. Renal function is normal. Blood counts remain correct. The patient’s weight and height remain however, below 3 pc. For this reason, he is also under care of an endocrinologist (Table 3).

## 4. Discussion

To this date, 34 patients with TALDO have been described [1]. The first case was reported in 2001 [2]. The clinical picture of TALDO includes cardiovascular disorders, renal function disorders, growth disorders, endocrine disorders, skin abnormalities, and the most common—liver dysfunction and haematological disorders. Intellectual development in patients with TALDO remains normal.

There is no specific treatment for TALDO. It is based on symptomatic therapy and include, among others, correction of coagulation disorders, albumin supplementation, transfusion of blood products, banding of oesophagal varices (in patients with end-stage liver disease). Patients remain under multi-specialist care. Experimental therapy with NAC (N-acetylcysteine) was proposed, which, according to the authors, is important in reducing the concentration of AFP in patients with TALDO [7]. However, a reduction in AFP was also observed in patients without this therapy [6,8].

Only 2 patients with TALDO undergoing liver transplantation have been described [3,4]. End-stage liver disease was observed in both patients and cirrhosis and hepatocarcinoma (HCC) were reported in one of them. Both patients had liver transplantation at the age of one, before being diagnosed with TALDO.According to the literature both patients after long-term observation, 7 and 3 years after LT respectively, present normal liver function and blood counts. Williams also reported (based on personal communication) about one patient transplanted at 5 months of age from a deceased donor, but this child died two weeks after LT because of lung infection and respiratory failure [1].

We present cases of two brothers whose diagnosis of TALDO was made long before LT (at 3 years of age and at 6 months of age, respectively). As the cause of liver dysfunction was known before the decision of LT qualification, which was particularly difficult and delayed in both children, especially that in analysis of the previously published cases of patients with TALDO we did ot find reports on the long-term follow-up of transplanted patients.

Liver transplantation in patients with TALDO have been associated with an uncertain prognosis and the risk of disease recurrence. Due to the multi-organ nature of the disease, the LT procedure was burdened with a high risk of intra-and postoperative complications, including graft failure. There was a high risk of progressive renal failure as shown by the observations of patients with TALDO [9]. Finally Patient I was qualified to LT due to the suspicion of HCC development, while Patient II was qualified due toend-stage liver disease.

In a long-term, almost 2-year follow-up, the graft function is normal in both our patients. No coagulation disorders are observed in any of the patients. Serum albumin levels remain normal. Tubulopathy is the most common manifestation of kidney injury in TALDO patients. The first symptoms are proteinuria, tubular acidosis, aminoaciduria, and glycosuria [6]. Chronic kidney disease develops gradually [9]. In Patient I periodically discreetly elevated concentrations of creatinine and cystatin C are observed. Patient II maintain normal concentrations of creatinine and cystatin C. The ultrasound examination of the kidneys shows abnormalities in both brothers. They receive nephroprotective treatment with Ramipril and we do not observe progression of renal function after LT.

Haematological abnormalities in patients with TALDO have many causes [10]. Splenomegaly, which occurs in almost half of patients with TALDO and hypersplenism, may be the cause of leukopenia and thrombocytopenia. Splenomegaly was not observed in long-term follow-up in our patients. The platelet count in both boys remain within the normal range. Patient I periodically has a mild leukopenia. In Patient II, the leukopenia episode 6 months after LTX was most likely the result of treatment side effects. After discontinuation of MMF treatment, parameters returned to normal. Anaemia in patients with TALDO occurs in approximately 77% of patients. Potential causes are bleeding, haemolysis, decreased haematopoiesis, and renal dysfunction. In Patient I, a mild degree of anaemia persists. In patient II, red cell parameters are normal which may reflect longer history of disease in older brother as well impaired renal function.

Despite the normal function of the graft, both patients have still significant growth deficiency. Patient I has malnutrition additionally. For this reason, both patients require extended endocrine diagnostics.

TALDO is a multi-organ disease, therefore LT does not cure patients from the disease. It is necessary to regularly evaluate several parameters, including haematological, renal, hepatic, pulmonary and coagulation in long-term follow-up. The assessment of endocrine abnormalities and observation of physical development and nutritional parameters are also extremely important. Despite positive experience in both our patients and another two patients after LT reported in the literature further observations are necessary to assess the real role of liver transplantation in the improvement of long term prognosis of patients with TALDO, of which ab. 25% die before achieving adult age on standard symptomatic care [1].

## 5. Conclusions

In summary, liver transplantation can be a therapeutic option in children with TALDO and should be considered in patients with severe chronic and end-stage liver disease. Long term follow-up is necessary to assess the impact of liver transplantation on quality of life, survival time and the course of the disease.

## Figures and Tables

**Table 1 children-08-00746-t001:** Clinical course of Patients I and II before LT.

	Patient I	Patient II
Gender	Male	Male
Genotype	Homozygotec.575G>A, p.Arg192His	Homozygotec.575G>A, p.Arg192His
Delivery, weeks	38	37
Birth weight, g	2380	2700
Birth lenght, cm	48	50
Age of diagnosis	3 yrs	5 mo
Hepatosplenomegaly	+	+
Liver function problems	+	+
Bleeding diathesis	+	+
Anemia	+	+
Thrombocytopenia	+	+
Cardiac abnormalities	-	PFO, First-degree AV Block
Developmental delay	-	-
Skin changes	+	+
Abnormal genitalia	Cryptorchidism	Cryptorchidism
Renal problems	tubulopathy at 5 yrs	tubulopathy at 4 yrs
Hepatocellular Ca	susp.due to nodular liver, not confirmed in histopathology	-

+ present, - absent, yrs—years, mo—months, PFO—patent foramen ovale, AV—atrioventricular, susp.—suspicion, Ca—carcinoma.

**Table 2 children-08-00746-t002:** Trasplantation details and postLT follow-up of Patients I and II.

	Patient I	Patient II
Age at listing for LT	11 yrs	8 yrs
Age of LT	14 yrs	11 yrs
Type of LT	Deceased donor, A to A group, whole liver	Deceased donor, A to A group, whole liver
Early complications of LT	epileptic episode	Intraabdominal bleeding, epileptic episode, HT, CMVinfection, AR susp.
Late complications of LT	anemia, leukopenia, kidney dysfunction	-
Age at last evaluation	16 yrs	13 yrs
Growth after LT	<3 pc	<3 pc

+ present, - absent, yrs—years, susp.—suspition, Ca—carcinoma, HT—hypertension, AR—acute rejection, LT—liver transplantation, pc—percentile.

**Table 3 children-08-00746-t003:** Clinical and biochemical data before and in post LT follow-up.

	Patient I					Patient II				
	1 Mo before LT	Discharge day after LT	6 Mo after LT	12 Mo after LT	18 Mo after LT	1 Mo before LT	Discharge day after LT	6 Mo after LT	12 Mo after LT	18 Mo after LT
Age (y and mo)	14 y 7 mo	14 y 8 mo	15 y 2 mo	15 y 8 mo	16 y 2 mo	11 y 10 mo	12 y 1 mo	12 y 7 mo	13 y 1 mo	13 y 6 mo
Height (cm)	139	139	143	144.5	146	119	119	122	123.5	129.2
Weight (kg)	30	29.2	32.5	32.5	34.5	19	21	22	23	29
BMI (kg/m^2^)	15.53	15.11	15.89	15.67	16.19	13.42	14.83	14.78	15.2	17.43
Hb (mg/dl)	11.2	11.0	11.3	12.2	12.7	9.8	12.6	11.8	12.1	14.2
WBC (10^3^/uL)	4.61	4.9	2.71	4.0	3.5	6.41	4.7	1.81	4.2	6.02
PLT (10^3^/uL)	97	330	169	204	203	116	179	201	216	208
INR	1.23	0.9	1.0	0.92	0.98	1.2	0.93	0.96	0.96	0.98
AST (U/L)	61	22	15	16	18	77	18	19	18	21
ALT (U/L)	38	13	10	12	16	31	21	13	12	12
GGTP (U/L)	102	53	9	11	10	50	107	14	13	13
Total bilirubin (mg/dL)	1.02	0.52	0.31	0.52	0.6	2.8	1.05	0.54	0.65	0.64
Direct bilirubin (mg/dL)	0.47	0.13	0.17	0.25	0.3	1.87	0.38	0.22	0.25	0.33
Albumin (g/L)	3.44	4.15	3.96	3.97	4.02	3.47	4.47	4.26	4.16	4.22
Creatinin (mg/dL)	0.92	1.01	1.06	1.19	1.0	0.33	0.33	0.63	0.53	0.65
eGFR (mL/min)	62.39	56.84	55.72	50.15	60.29	148.93	148.93	79.98	96.24	82.09
Cistatin C (mg/L)	1.19	1.63	1.43	1.37	1.38		1.61	1.23	1.28	1.18
AFP (IU/mL)	9.42						4.24			
EBV DNA	N.d.	N.d.	N.d.	N.d.	N.d.	N.d.	N.d.	N.d.	N.d.	N.d.
IMS		Tac+MMF +ST	Tac +MMF	Tac +MMF	Tac		Tac+MMF +ST	Tac+MMF+ST	Tac+ST	Tac+ST

Mo: months, y: years, LT: liver transplantation, BMI: Body Mass Index, eGFR: Glomerular Filtration Rate according to the Schwartz formula, N.d.: not detected, IMS: immunosuppression, Tac: tacrolimus, MMF: mycophenolate mofetil, ST: steroids.

## Data Availability

The data is not publicly available due to the protection of personal data.

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
