# Peer review of "Successful Liver Transplantation in Two Polish Brothers with Transaldolase Deficiency"

_children, 2021, doi:10.3390/children8090746_

Round 1

Reviewer 1 Report

This paper will for sure complement the knowledge in the field of rare metabolic diseases.

At line 92: could you please clarify the neurologic diagnosis "loss of logical contact"?

I suggest replacing the term "liver failure" with "end-stage liver disease" as per the most recent Baveno guideline (Baveno VI).

Thank you!

Author Response

Point 1: At line 92: could you please clarify the neurologic diagnosis "loss of logical contact"?

Response 1: Thank you for your comments and keen observations. The patient had a brief episode of confusion. He did not logically answer the questions asked. Neurological diagnostics showed no abnormalities. The episode resolved by itself. 

The sentence has now been corrected and reads as follows:

"The episode of a disorientation was noticed, but neurological examinations (including CNS imaging, EEG) did not confirm the organic and functional pathology of CNS."

Point 2: I suggest replacing the term "liver failure" with "end-stage liver disease" as per the most recent Baveno guideline (Baveno VI).

Response 2: Thanks for your comment. The text has been reviewed and corrected according to your suggestions.

Thanks for your comments!

Reviewer 2 Report

Authors have described Transaldolase deficiency cases requiring LT for ESLD. 

Introduction Line 33 and 34: suggest revising to transaldolase deficiency has varied presentation including .....

Line 74 oesophageal varices were revealed by "Esophagogastroscopy or EGD" and not just gastroscopy. 

Line 75 :suggest changing Organ's function decompensation to liver decompensation. 

Line 77 change to worsening synthetic function

LIne 79: . Two years later severe bleeding from oesophagal varices 80 led to a coma.--->IS this hepatic encephalopathy resulting in coma  or hypotension and brain injury? please clarify. 

Author Response

Point 1: Introduction Line 33 and 34: suggest revising to transaldolase deficiency has varied presentation including .....

Response 1: Thank you for your comments. The sentence has now been corrected and reads as follows:

"Transaldolase deficiency has varied presentation including end-stage liver disease, renal tubular dysfunction, coagulopathy, anemia,trombocytopenia, congenital cardiac abnormalities and hormonal disorders. Bone mineralization disorders and short stature are found in patients with TALDO."

Point 2: Line 74 oesophageal varices were revealed by "Esophagogastroscopy or EGD" and not just gastroscopy. 

Response 2: Thank you for your comments. The sentence has now been corrected and reads as follows:

"The nodular liver fibrosis was detected by ultrasonography and oesophageal varices were revealed by esophagogastroscopy."

Point 3: Line 75 :suggest changing Organ's function decompensation to liver decompensation. 

Response 3: Thank you for your comments. The sentence has been amended and now reads as follows:

"Progression of liver damage led to liver decompensation at the age of 5 yrs, with massive ascites and peripheral oedema (scrotum and eyelids)."

Point 4: Line 77 change to worsening synthetic function

Response 4: Thank you for your comments. The sentence has now been corrected and reads as follows:

"Laboratory tests revealed worsening synthetic function of the liver with hypoalbuminemia, coagulopathy and hyperaminotransferasemia."

Point 5: Line 79: . Two years later severe bleeding from oesophagal varices 80 led to a coma.--->IS this hepatic encephalopathy resulting in coma  or hypotension and brain injury? please clarify. 

Response 5: Thank you for your comments and keen observations. The coma was the result of a hypovolemic shock induced by bleeding esophageal varices.The sentence has now been corrected and reads as follows:

"Two years later severe bleeding from oesophagal varices led to hypotension and hypovolemic shock resulting in coma."

Thank you for your comments!